


# Temporally-resolved sectoral and regional contributions to air pollution in Beijing: Informing short-term emission controls

Tabish Umar Ansari[1], Oliver Wild[1], Edmund Ryan[2], Ying Chen[1], Jie Li[3], and Zifa Wang[3]

[1]Lancaster Environment Centre, Lancaster University, Lancaster, UK
[2]School of Mathematics, University of Manchester, Manchester, UK
[3]State Key Laboratory of Atmospheric Boundary Layer Physics and Atmospheric Chemistry, Institute of Atmospheric Physics, Chinese Academy of Sciences, Beijing, China

*Correspondence to:* Tabish Umar Ansari (t.ansari@lancaster.ac.uk) and Oliver Wild (o.wild@lancaster.ac.uk)

**Abstract.** We investigate the contributions of local and regional emission sources to air quality in Beijing to inform the design of short-term emission control strategies for mitigating major pollution episodes. We use a well-evaluated version of the WRF-Chem model at 3 km horizontal resolution to determine the daily accumulation of pollution over Beijing from local and regional sources in October 2014 under a range of meteorological conditions. Considering feasible emission reductions

across residential, transport, power and industrial sectors, we find that one-day controls on local emissions have an immediate effect on $PM_{2.5}$ concentrations on the same day, but can have lingering effects as much as five days later under stagnant conditions. One-day controls in surrounding provinces have the greatest effect in Beijing on the day following the controls, but may have negligible effects under northwesterly winds when local emissions dominate. To explore the contribution of different emission sectors and regions, we perform simulations with each source removed in turn. We find that residential and

industrial sectors from neighbouring provinces dominate $PM_{2.5}$ levels in Beijing during major pollution episodes, but that local residential emissions and industrial/residential emissions from more distant provinces can also contribute significantly during some episodes. We then perform a structured set of perturbed emission simulations to allow us to build statistical emulators that represent the relationships between emission sources and air pollution in Beijing over the period. We use these computationally fast emulators to determine the sensitivity of $PM_{2.5}$ concentrations to different emission sources and the interactions between

them, including for secondary PM, and to create pollutant response surfaces for daily average $PM_{2.5}$ concentrations in Beijing. We use these surfaces to identify the short-term emission controls needed to meet the national air quality target of daily average $PM_{2.5}$ less than $75\,\mu g\,m^{-3}$ for pollution episodes of different intensities. We find that for heavily polluted days with daily mean $PM_{2.5}$ higher than $225\,\mu g\,m^{-3}$, even emission reductions of 90% across all sectors over Beijing and surrounding provinces may be insufficient to meet the national air quality standards. These results highlight the regional nature of PM pollution and

the challenges of tackling it during major pollution episodes.

## 1 Introduction

Beijing, located at the foot of the Yanshan mountains on the northern edge of the heavily-populated North China Plain, has consistently been named among the most polluted capital cities in the world (State of Global Air, 2019; Global burden of





disease, 2016). Key air pollutants including carbon monoxide, sulphur dioxide, nitrogen oxides, ozone and $PM_{2.5}$ (particulate matter with diameter less than $2.5\,\mu m$) often breach the World Health Organization (WHO) and national air quality standards, posing risks to human health, visibility and climate (Lelieveld et al., 2015; Luan et al., 2018; UNEP and WMO, 2011). To address the severe ambient air pollution in Chinese cities, the State Council of China launched the Air Pollution Prevention

and Control Action Plan in 2013 setting targets to reduce $PM_{2.5}$ concentrations over the Beijing–Tianjin–Hebei region by 25% from their 2013 levels by 2017, and to reduce annual mean $PM_{2.5}$ concentrations in Beijing to $60\,\mu g\,m^{-3}$ (Zheng et al., 2018; Wei et al., 2017). The mitigation strategies focused on long-term measures such as gradual phase-out of residential biofuel use, changes in industrial technology, improved vehicle fuel standards and relocation of coal-fired power plants. The air quality in Beijing improved substantially during 2013–2017, and the annual mean $PM_{2.5}$ concentration decreased from $89.5\,\mu g\,m^{-3}$ in

2013 to $58\,\mu g\,m^{-3}$ in 2017, largely due to the implementation of these long-term local and regional emission control measures (Ma et al., 2018; Cheng et al., 2019; Zheng et al., 2018).

While recent long-term emission reductions have brought substantial improvements in overall air quality, the region continues to experience frequent haze episodes characterized by high levels of $PM_{2.5}$, particularly in winter (Dang and Liao, 2019; Xiao et al., 2020). Additional emergency measures generally lasting 3–7 days are necessary to prevent these extreme pollution

episodes, especially under stable meteorological conditions that are conducive to formation and accumulation of very high levels of particulate matter. Such short-term emission controls have been tested, with some success, during special events such as the 2008 Beijing Olympics, the Asia-Pacific Economic Cooperation (APEC) summit in 2014 and the China Victory Day Parade in 2015 (Zhang et al., 2015; Xu et al., 2019, 2017). However, the success of these controls has often been greatly aided by favourable weather conditions (Liu et al., 2017; Liang et al., 2017; Gao et al., 2017) and the same control strategies are liable

to fail under different meteorological conditions. For example, while national air quality standards were met during the APEC summit, we have shown that they would have been greatly exceeded under the same emission controls had the summit been held two weeks earlier when the weather was less favourable (Ansari et al., 2019). Previous studies have advocated application of emission controls over a wider geographical region (Guo et al., 2016; Wen et al., 2016; Ansari et al., 2019) but have not identified the spatial or temporal scales needed for successful policy implementation, or proposed any general framework to

devise future mitigation strategies that account for differing meteorological conditions. In this study we use a range of new modelling approaches, including one-day emission reductions and statistical emulation, to gain a detailed understanding of the magnitude and timing of local and regional source contributions to $PM_{2.5}$ concentrations in Beijing under varying meteorological conditions. We generate pollutant response surfaces based on these different emission sectors and regions, and demonstrate how they can be used to guide the development of future short-term emission control policies in the city.

## 2   Modelling approaches and motivation

We use the Weather Research and Forecasting model with Chemistry (WRF-Chem) version 3.7.1 at a horizontal resolution of $27\,km$ over China with nested domains over Northern China at $9\,km$ and the North China Plain at $3\,km$ resolution. Gas-phase chemistry in the model is represented by the Carbon Bond mechanism version Z (CBMZ) which is coupled with the Model





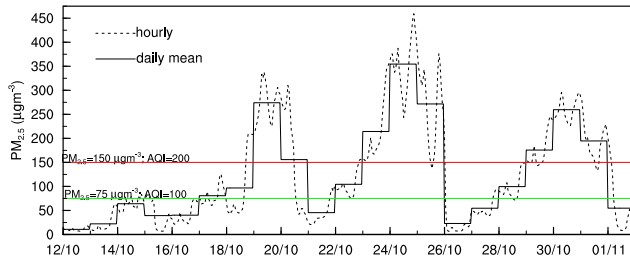

**Figure 1.** Average simulated hourly and daily mean $PM_{2.5}$ in Beijing in October 2014. The national air quality standard for 24 hr average $PM_{2.5}$ of $75\,\mu g\,m^{-3}$ is shown in green, and the threshold for heavy pollution, $150\,\mu g\,m^{-3}$, is shown in red.

for Simulating Aerosol Interactions and Chemistry (MOSAIC) aerosol module with eight aerosol size bins (Zaveri and Peters, 1999; Zaveri et al., 2008). We use meteorological fields from the European Centre for Medium-Range Weather Forecasts (ECMWF), and anthropogenic emissions from the Multi-resolution Emission Inventory for China (MEIC) appropriate for 2014. Further details of the model configuration and evaluation over North China are provided in a previous study where

we explored emission controls in Beijing during the Asia-Pacific Economic Cooperation (APEC) summit in November 2014 (Ansari et al., 2019). In this previous study we demonstrated that the model can reproduce the magnitude and variation of key pollutants over Beijing well, and showed that meeting national air quality standards over the period was critically dependent on the weather conditions at the time. Formulation of effective short-term emission control policies therefore needs to account for the important role played by meteorological processes. In this study we focus on the same October-November period

in 2014 and investigate the key source sectors and regions responsible for short-term pollution episodes during a range of meteorological conditions.

Figure 1 shows hourly and daily mean simulated $PM_{2.5}$ concentrations for Beijing at the end of October 2014, the period just before emission controls were implemented for APEC. In the last 15 days of October, only three days met the daily national Class 2 air quality standard of 24 h average $PM_{2.5}$ concentration less than $75\,\mu g\,m^{-3}$ (Air Quality Index, AQI=100), and

eight days exceeded the higher threshold for heavy pollution of $150\,\mu g\,m^{-3}$ (AQI=200). Reducing the 24 h average $PM_{2.5}$ concentration below $75\,\mu g\,m^{-3}$ over the entire period would require strict, carefully tailored emission reductions across multiple sectors and regions that are timed to provide maximum benefit during the heaviest pollution episodes.

For APEC, emission controls were implemented in two stages: an initial phase (APEC 1) that covered Beijing and some western districts of Hebei province including Baoding, Langfang, Shijiazhuang, Xingtai and Handan, and a second more

stringent phase (APEC 2) that additionally covered Tianjin, Tangshan, Cangzhou, Hengshui, Dezhou, Binzhou, Dongying, Jinan, Liaocheng and Zibo. In this study we consider three broad regions of control: local emissions from Beijing, Near-Neighbourhood emissions from the North China Plain and Far-Neighbourhood emissions from surrounding provinces (see Figure 2). The Near-Neighbourhood region used here is defined to match the APEC 2 control region described above. Since we have shown that the APEC controls applied over this period were insufficient to meet healthy air quality standards (Ansari et al.,

2019), we consider a Far-Neighbourhood region which covers parts of Shanxi, Shaanxi, Shandong, Henan, Inner Mongolia,





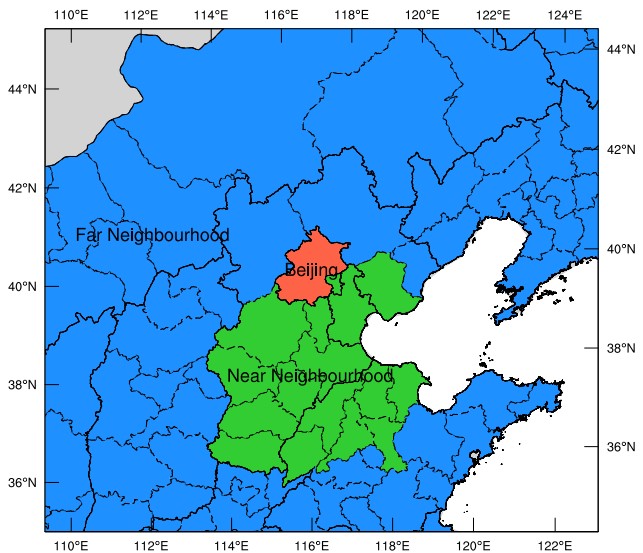

**Figure 2.** North China (model domain 2) showing the emission control regions used in this study.

Liaoning and Jilin provinces. This covers most of our second model domain, as shown in figure 2; the outer model domain (not shown) covers the whole of China.

A number of measurement and modelling studies have investigated source apportionment of air pollution in Beijing. Most measurement studies have used the Positive Matrix Factorization (PMF) technique which effectively attributes $PM_{2.5}$ to dif-
ferent emission sectors but does not provide the district-based regional contributions critical for policymaking (Zhang et al., 2013; Zíková et al., 2016; Shang et al., 2018). Li et al. (2014) developed a tagging approach using the Nested Air Quality Prediction Modelling System (NAQPMS) which apportions primary aerosols to their emission sources and secondary aerosols to their geographic region of formation. However, from the perspective of control policy formulation for secondary aerosol-dominated episodes, it would be more valuable to apportion secondary aerosols to specific precursor emission sources rather
than their region of formation. Li et al. (2015) performed a comprehensive modelling study using the Particulate Matter Source Apportionment Technology (PSAT) emissions-tagging approach in the Comprehensive Air Quality Model with Extensions (CAMx) at a relatively coarse resolution of 36 km to derive district-based source contributions. However, this approach does not capture the interactions between secondary PM precursor species which may be particularly important during episodes with high nitrate levels (Burr and Zhang, 2011). Recently, Chen et al. (2019) used the PSAT approach at 12 km resolution to
calculate source contributions from districts included in the Chinese government's "2+26" regional emission control strategy, although they did not include the effect of surrounding regions in their analysis, which may be needed to formulate successful emission control policy under less favourable meteorological conditions. Moreover, none of these studies provide information on the temporal contribution of different sources which is important for the design of short-term emergency control measures. Matsui et al. (2009) performed a simple exploration of the temporal nature of source contributions from different regions. They
found that primary aerosols in Beijing are affected by emissions within 100 km in the previous 24 h and secondary aerosols





over Beijing are affected by emission sources within 500 km in the previous 3 days. However, a more detailed understanding of the temporal nature of regionally-resolved sectoral source contributions under a range of meteorological conditions is required to aid robust short-term emission control policymaking for Beijing.

With the aim of informing short-term emission reduction policies, we investigate the temporal contributions from different
emission sectors and regions by conducting a number of different simulation experiments. In section 3 we determine the contribution from each day of anthropogenic emissions over each major emission region to the temporal evolution of pollutant levels in Beijing. We investigate the total contribution from each sector and region in section 4. In section 5 we then explore the sensitivity of pollutant levels in Beijing to all sectors and regions simultaneously using Gaussian Process emulation, and build pollutant response surfaces for Beijing that can be used to determine the emission reductions needed to meet specified air
quality standards.

## 3   Temporal response to emission controls

To ensure good air quality, regional short-term emission controls have been implemented for high-profile events in Beijing over the past decade, including the 2008 Olympics, the 2014 APEC meeting, and the 2015 China Victory Day parade. These controls resulted in improved air quality, although for all three events the successful outcomes were greatly aided by favourable
meteorological conditions (Yang et al., 2011; Liang et al., 2017). In each case, emission controls were initiated substantially before the event: 18 days before for the Olympic games in 2008 (Gao et al., 2011; Yang et al., 2011), 7 days before for the APEC meeting in 2014 (Zhang et al., 2016; Sun et al., 2016) and 14 days before for the Victory Parade in 2015 (Liang et al., 2017; Zhao et al., 2017b). This period allowed emission controls to take effect and existing pollution to be swept away, but the optimal timing for controls that balances improvements in air quality against the economic and social costs of implementing them
remains unclear. To resolve this, it is important to understand how pollution from previous days and different regions builds up under different meteorological conditions and to determine the persistence of pollution from a single day of emissions.

We consider three regions of control: Beijing, Near-Neighbourhood and Far-Neighbourhood (see Fig 2) and conduct 60 five-day long model runs, 20 for each region, with emission reductions for the first day of the simulation only and standard baseline emissions for subsequent days (see Figure S1). We apply emission reductions following those implemented during the APEC
summit period: 40–50% for Beijing and 30–35% for surrounding districts (see Ansari et al., 2019). The period of interest was selected to cover a range of different meteorological conditions and includes a polluted period in mid to late October (14 days) and a cleaner period in early to mid November (also 14 days in length). The difference in simulated $PM_{2.5}$ concentrations between each model run and the baseline run gives the contribution from sources on a specific day over subsequent days.

Figure 3 shows the contribution of one-day emission changes applied across all three regions to $PM_{2.5}$ in Beijing for 21–27
October 2014. While emission reductions on a given day make a substantial contribution to $PM_{2.5}$ on that day, the benefits often extend to a number of subsequent days due to the meteorological conditions and timescales for transport. The most polluted days (e.g., 24 and 25 October) show accumulated contributions to $PM_{2.5}$ from as much as five days previously. Under clean, northwesterly winds, the pollution is swept away over the North China Plain, and the resulting clean days are dominated





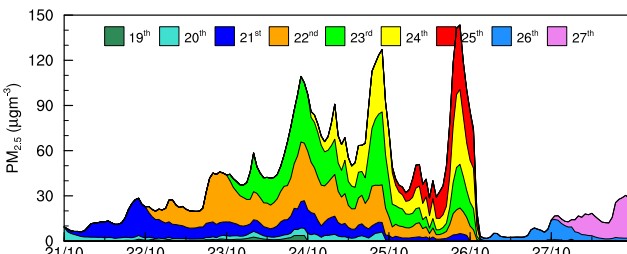

**Figure 3.** Contribution of successive days of emissions to $PM_{2.5}$ concentrations in Beijing for 21–27 October 2014.

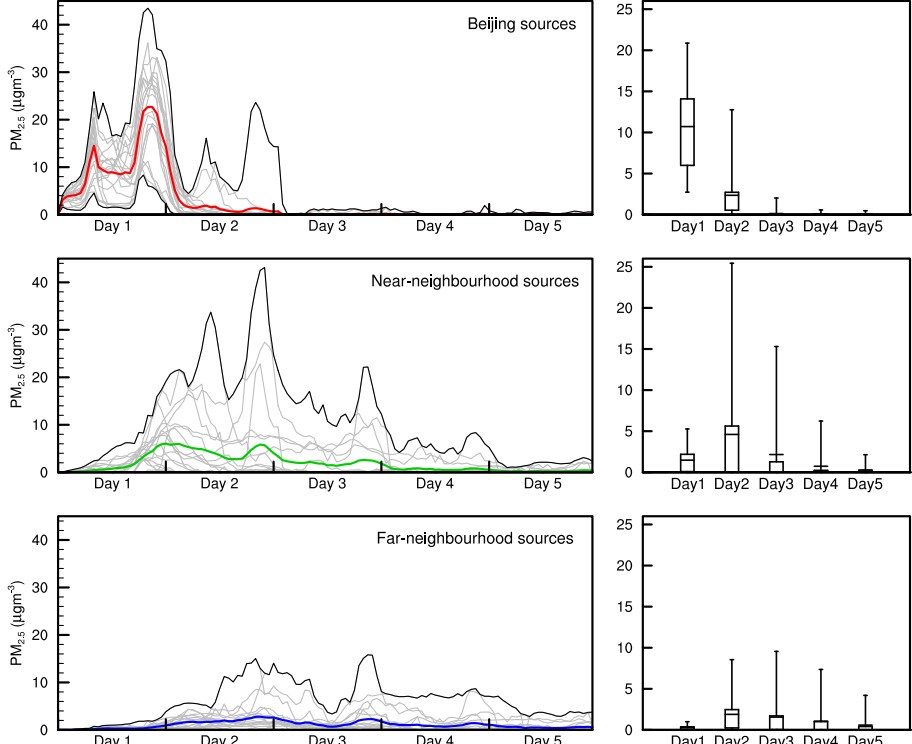

**Figure 4.** Time evolution of hourly (left) and daily (right) contributions to $PM_{2.5}$ concentrations in Beijing due to one-day emission changes in the Beijing (top), Near-Neighbourhood (middle) and Far-Neighbourhood (bottom) source regions in October/November 2014. The contribution of individual days is shown in grey, mean contributions are coloured, and maximum and minimum contributions are shown in black. Boxplots (right) show the mean, 25th/75th percentiles, and maximum/minimum contributions across all runs each day.

by fresh pollution from the same day, as seen on 26 October. The daily contributions vary substantially in magnitude as well as in timing, and pollution levels are typically higher at nighttime when the planetary boundary layer (PBL) is shallower, allowing local pollutants to build up.



To gain insight into how meteorological processes affect the timescales for transport from different regions, we explore the contributions from Beijing, Near-Neighbourhood and Far-Neighbourhood regions separately. Figure 4 presents the contributions to $PM_{2.5}$ concentrations in Beijing from one-day emission reductions relative to the day of control for the three different source regions over the 28-day period considered here. Contributions from Beijing sources affect $PM_{2.5}$ in Beijing immedi-

ately and typically last 1–2 days, with peak contributions ranging from 8–43 $\mu$g m$^{-3}$ (median value 21 $\mu$g m$^{-3}$) and occurring towards the end of the day of emissions control. Contributions from the near-neighbourhood region are somewhat delayed and typically extend until the third day. The contributions are generally smaller than those from Beijing sources and are greatest on the second day. Contributions from the far-neighbourhood region are more spread out and are further delayed in time, with very little contribution over Beijing on the day of control. The contributions are greatest in the two days following controls,

reflecting the timescales for transport, and can persist for as much as five days. The right hand panels in figure 4 show 24 h average contributions from each of the three source regions and highlight the variability in the magnitude and duration of the influence from a given day of emissions that is driven by the prevailing meteorological conditions. The highest hourly contribution from local controls in Beijing (43 $\mu$g m$^{-3}$) occurs for emissions on 25 October, which is one of the most polluted days during the study period (see Fig 1). The highest hourly contribution from near-neighbourhood controls is similar in magnitude

(also 43 $\mu$g m$^{-3}$) and occurs from emissions on 23 October which are trapped over Beijing during the pollution episode of 24 and 25 October. Similarly, the highest contribution from far-neighbourhood pulses occurs during the same pollution episode and arises from emissions one day earlier, on 22 October.

To characterize the source contributions for each simulation, we calculate the magnitude and timing of the peak contribution, the integrated contribution and its duration (see Table S1 in supplementary material). During the polluted period in October,

the highest integrated contribution comes from Beijing sources on five of the ten days, from near-neighbourhood sources on four days and from far-neighbourhood sources on one day. During the cleaner period in November, the highest integrated contribution is from Beijing sources on almost all occasions (nine out of ten days), and far-neighbourhood sources make a greater contribution than near-neighbourhood sources on eight of the days. Local Beijing sources are responsible for the highest peak contributions on six of the days in October and for all ten days in November. However near-neighbourhood sources also

make a major contribution on polluted days. Far-neighbourhood sources showed higher contributions than near-neighbourhood sources for seven of the days in November, but these were substantially smaller than contributions from Beijing sources. The effect of Beijing sources rarely extends to the end of the second day, whereas the contributions from more distant sources can extend to almost five days. The greatest contributions from all regions occur in late afternoon or early evening in Beijing, reflecting the accumulation of pollutants in the more stable evening boundary layer, along with accumulation from rush hour

traffic sources.

While the temporal contributions from each region show distinct mean behaviour, they also show significant variability associated with the differing meteorological conditions each day. To examine the effect of meteorology, we investigate the relationship between peak and integrated contributions and meteorological parameters such as daily average wind speed and direction. Some key relationships are shown in Figure 5. We find that peak contributions from Beijing sources show a clear

negative correlation (r value of -0.74) with average wind speed in Beijing on the day of the emissions. This is as expected, and





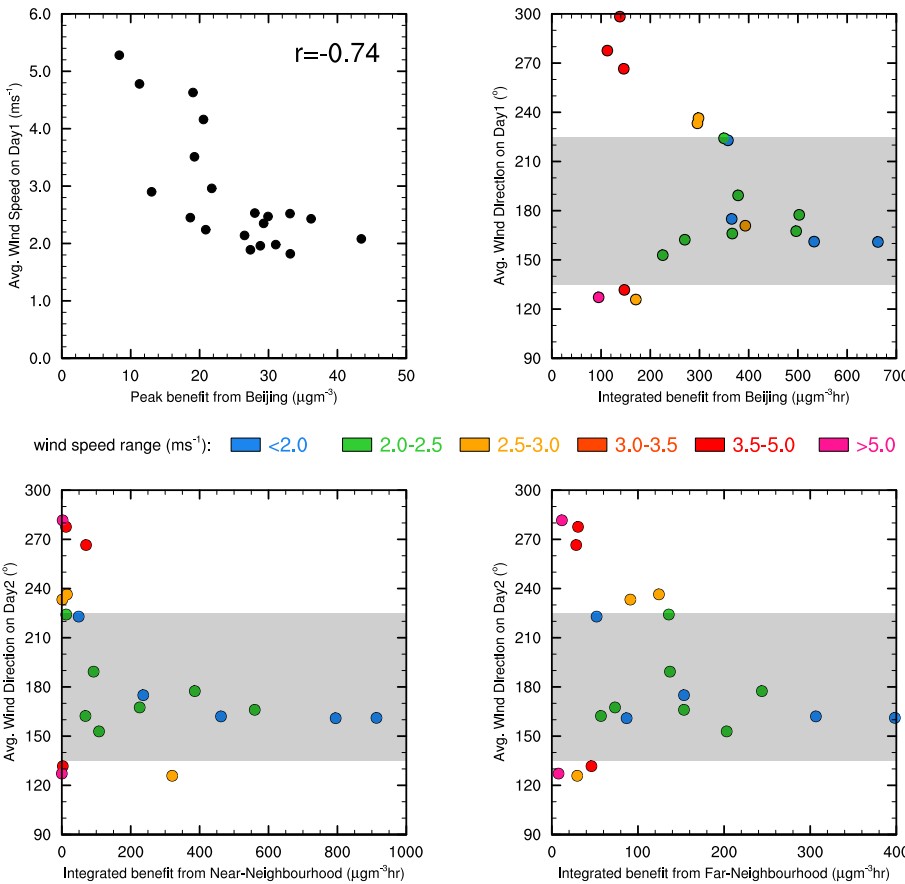

**Figure 5.** Effect of meteorology on peak and integrated contributions from the three regions of control. (a) Peak contribution from Beijing sources vs average wind speed on day 1, (b) integrated contribution from Beijing vs average wind direction on day 1, and integrated contribution from (c) Near-Neighbourhood and (d) Far-neighbourhood sources vs average wind direction on day 2. The shaded area highlights southerly winds.

reflects the greater build up of pollutants under more stagnant conditions. The wind speed on subsequent days does not show a clear relationship with the contributions from any of the sources. The integrated contribution from all three source regions is higher when the wind is blowing from the south. This reflects the fact that the strongest regional emission sources lie over the North China Plain south of Beijing. Higher integrated contributions were seen during southerly winds when the wind speed
5 was lower, demonstrating the importance of the build up of pollutants from across the North China Plain under more stable, anticyclonic conditions. Winds from other sectors were generally stronger and led to very little contribution from regional sources over Beijing.

Overall, it is clear that local emission controls in Beijing provide an immediate but typically relatively short-lived benefit to air quality in the city, with the magnitude of the benefit dependent on whether stagnant meteorological conditions with low
10 wind speed are prevailing. Regional emission controls over the North China Plain generally lead to smaller benefits which





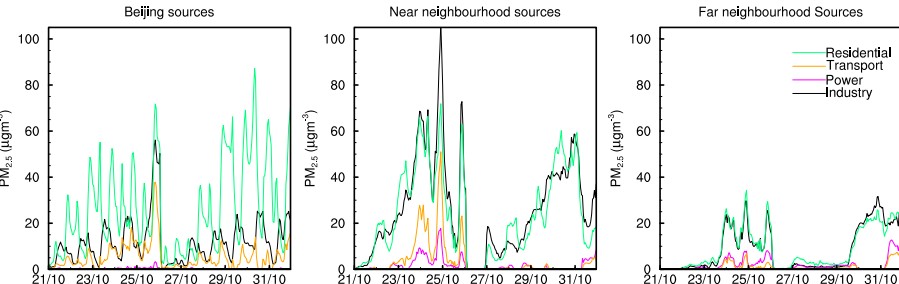

**Figure 6.** Contribution of residential, transport, power and industry emission sectors from local, near-neighbourhood and far-neighbourhood source regions to $PM_{2.5}$ concentrations in Beijing during 21–31 October 2014.

start later and last longer, although their contribution may rival those from local emissions under weak, southerly winds. More distant regions may also make a contribution under these conditions, although these typically take a number of days to build up. These results suggest that mitigating major pollution episodes in Beijing requires control of local emissions one or two days in advance, control of near-neighbourhood emissions two to three days in advance, and control of far-neighbourhood emissions

three to four days in advance.

## 4    Sectoral emission controls

An understanding of the contribution from individual sectors (industry, transport, residential and power generation) from different regions is also important to inform successful emission control policies. To determine the impact of each source sector in each different region, we conduct model simulations where each of the twelve emission sources (four sectors over three

regions) was removed in turn for the period of 21–31 October. The difference between each run and the baseline run yields an estimate of the contributions of the source considered. We conducted an additional run with all twelve emission sources removed to provide an estimate of the background contribution from sources outside the regions considered here.

The October simulation period is characterised by two distinct pollution episodes: 21–25 October (episode 1) and 26–30 October (episode 2). During each episode the total $PM_{2.5}$ concentration over Beijing builds up over a five-day period

and peaks on the fifth day. Figure 6 shows hourly time-series of contributions from each emission source. In general, residential and industrial sectors make much greater contributions than the power and transport sectors. The contribution of Beijing emission sources shows a more pronounced diurnal variation than that of the regional sources. This is because the Beijing plume is still fresh and contains the diurnal signal of local emissions whereas regional plumes are transported to Beijing over longer distances and are better mixed, losing this diurnal signal. All sources show a build-up during the two episodes, but the

rate of this increase is high for Beijing sources, low for near-neighbourhood sources and lower still for far-neighbourhood sources. Far-neighbourhood contributions show a delayed onset for this increase compared with those from nearer sources, highlighting that 3–4 days of stagnation over Beijing are needed before Far-Neighbourhood sources start to make a substantial contribution. This is consistent with our findings on the temporal response to emission changes in Section 3. It is interesting to





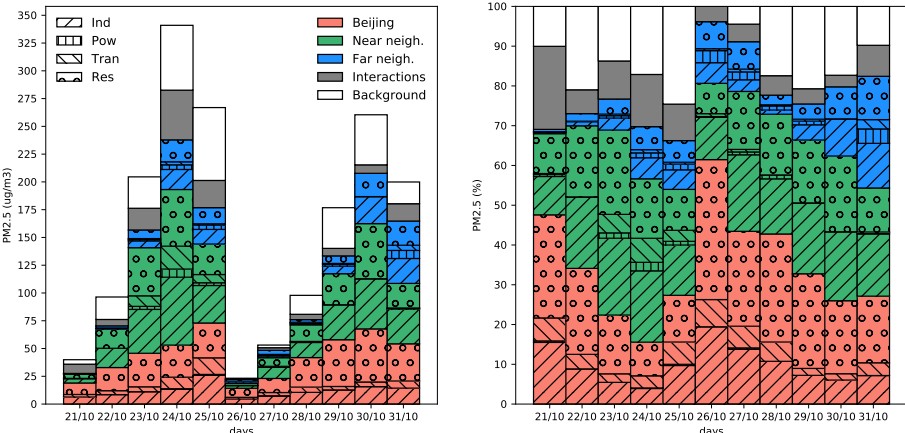

**Figure 7.** Absolute (left) and relative (right) contributions to daily mean $PM_{2.5}$ in Beijing for 21–31 October obtained by removing one emission sector from one region at a time. Contributions from the three regions are shown in different colours and from the four sectors are shown with distinct patterns. Contributions from other sources are shown in white and contributions due to interactions between emissions from different sectors are shown in grey.

note that the source contributions do not peak at the same time, and therefore the maximum concentrations of $PM_{2.5}$ do not reflect the maximum contributions from each source. Figure 6 reveals the anatomy of the two $PM_{2.5}$ episodes shown in Fig 1 and shows that episode 1 was dominated by regional pollution, particularly industrial, residential and transport emissions from near-neighbourhood sources, while episode 2 had greater influence from local residential emissions which exceeded the con-

5 tributions from regional emissions. This is because episode 1 experienced stronger southwesterly winds during 23–24 October ($4\,\mathrm{m\,s^{-1}}$) that brought $PM_{2.5}$ from high-emission sources south of Beijing enabling greater accumulation over the city before it was swept away by the clean northeasterly winds the next day. Episode 2 experienced weak southeasterly winds during 29–30 October ($2\,\mathrm{m\,s^{-1}}$) which did not bring as much $PM_{2.5}$ from neighbouring sources but allowed local emissions to build up under calmer conditions. See Fig S2 in the supplement for more details of the meteorological conditions during this period.

Figure 7 presents absolute and relative contributions to daily mean $PM_{2.5}$ in Beijing from the 12 emission sources along with those from background sources. The remaining difference between the sum of these contributions and the baseline $PM_{2.5}$ concentrations is attributed to interactions between emissions from different sources. The largest contributions are from Beijing and Near-Neighbourhood sources on most days, which average 35% and 32% respectively. However, Far-Neighbourhood sources make a substantial contribution (up to 28%) on polluted days. The two episodes have different characteristics, as

noted above, with the first episode more greatly affected by near-neighbourhood emissions while the second showed a greater influence from local emissions, notably from the residential sector. The spatial distribution of 24 h average $PM_{2.5}$ for the two days is shown in Fig 8, which reveals the differing extents of the episodes. Elevated $PM_{2.5}$ concentrations were present over a much wider area during the first episode, and the contributions from near-neighbourhood sources were substantially higher. The differences between these two pollution episodes just five days apart emphasises the importance of meteorological processes in



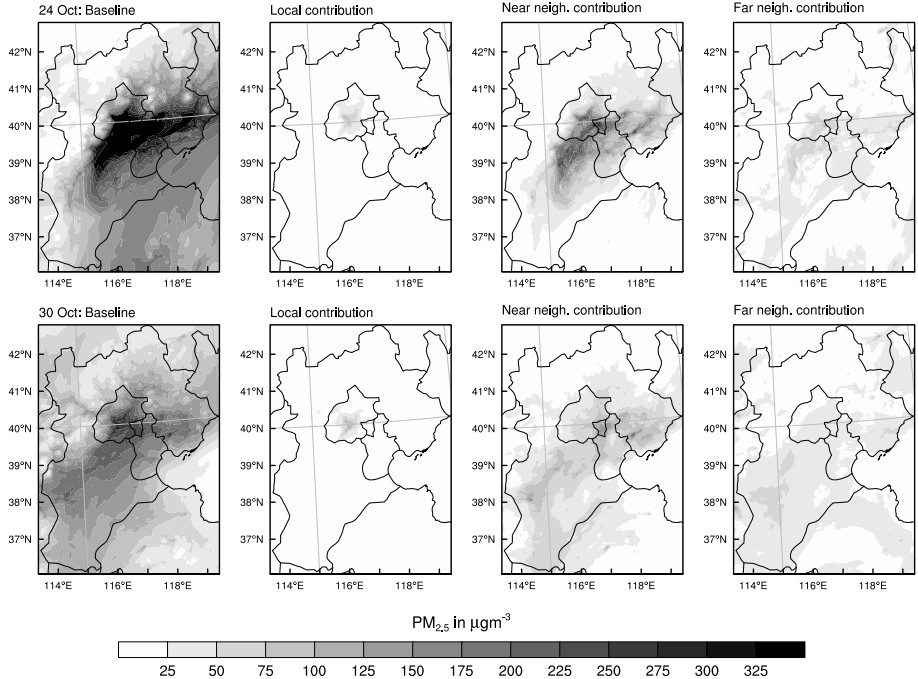

**Figure 8.** Daily mean $PM_{2.5}$ concentrations over the study region for 24 October (top) and 30 October (bottom) pollution episodes. Total $PM_{2.5}$ concentrations are shown alongside individual contributions from local, near-neighbourhood and far-neighbourhood regions derived from one-at-a-time zero-out sensitivity simulations.

governing the contributions of different sources. Note that the baseline concentration fields shown in the leftmost panels also include contributions from background and natural sources as well as from local, near-neighbourhood and far-neighbourhood emissions before the start of simulations.

The contributions of key sources build up consistently over the course of each pollution episode. A sizeable portion of
$PM_{2.5}$ is attributed to interactions between gas-phase precursors from different sources, and this reflects non-linear processes, particularly the formation of secondary aerosols (Zhao et al., 2017a). In general, days with higher contributions from transport and power sectors show the greatest interactions. These sectors are major sources of nitrogen dioxide ($NO_2$) and sulphur dioxide ($SO_2$) respectively and this nonlinear behaviour can be explained by competition between $SO_2$ and $NO_2$ in the presence of ammonia ($NH_3$) to form sulphate and nitrate aerosols. A reduction in power sector emissions alone reduces $SO_2$ and
therefore ammonium sulphate particles (($NH_4$)$_2SO_4$), thereby freeing up $NH_3$ gas which may then react with $NO_2$ to form ammonium nitrate particles ($NH_4NO_3$). This leads to little change in total $PM_{2.5}$ concentrations, offsetting the benefits of the emission reduction. Reducing emissions from both power and transportation sectors reduces the $NO_2$ available to react with the freed $NH_3$, leading to more efficient $PM_{2.5}$ reduction.

To place the sectoral and temporal attribution approaches used here into context, we compare the cumulative contributions
from one-day emission controls across all sectors with those from continuous emission controls on individual sectors. We





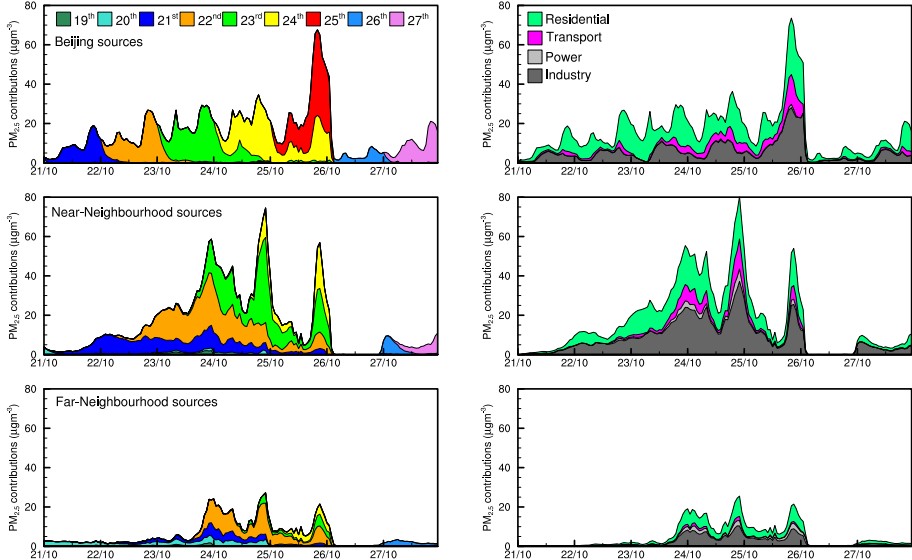

**Figure 9.** Contributions of one-day emission controls (left) and continuous sectoral controls (right) to $PM_{2.5}$ in Beijing showing how $PM_{2.5}$ can be attributed to the day, region and sector of emissions. Contributions from each sector (derived at 100%) were scaled down to match the 30–50% reductions used for the one-day controls.

scale the contributions from individual sectors by 45% for Beijing sources and by 35% for regional sources to match the APEC-like emission reductions applied in the one-day control runs described in Section 3. A comparison of the cumulative contributions to $PM_{2.5}$ in Beijing from successive one-day pulses and from different emission sectors is shown for each source region in Fig 9. Local sources predominantly affect $PM_{2.5}$ on the day of emission, although they may have a substantial

effect on the second day under polluted conditions, as is evident from emissions on 24 October. Sources over neighbouring provinces have a more prolonged effect, and there may be substantial contributions from as many as four days previously during polluted episodes such as on the evening of 25 October. On clean days, such as 26 October, the contribution of these sources is negligible. A sectoral analysis shows that residential and industrial sectors make the biggest contribution to $PM_{2.5}$ in Beijing from local sources, with the former typically dominating during the evenings when pollutant levels peak. $PM_{2.5}$

from neighbouring provinces is dominated by industrial sources, and while residential sources are also important, there is also a significant contribution from transportation sources during the pollution episode when air arrives in Beijing from the North China Plain. The cumulative contributions of different sources based on one-day emission controls are very similar to those from the continuous sectoral controls, see Fig S3, demonstrating that the two approaches give comparable responses. This lends confidence in our independent attribution of the temporal and sectoral contributions to $PM_{2.5}$ in Beijing and suggests

that secondary particle production associated with interactions between precursors from different sectors and regions plays a relatively small role.



## 5 Combined source contributions

Assessing the combined effect of different emission changes and the nonlinear response of secondary aerosol to precursor emissions from different sources requires that emission changes are applied across a number of sources at the same time. However, exploring all combinations of emission regions and sectors considered here would be computationally prohibitive.

We therefore apply a computationally fast surrogate model to fully explore these relationships in a computationally tractable manner. We choose a Gaussian Process based statistical emulator as a surrogate model as it requires relatively few training runs and is well-suited to mapping the non-linear but continuous relationship between emissions and concentrations characteristic of $PM_{2.5}$ (see Ryan et al., 2018; Chen et al., 2020). We focus here on 12 emission sources that include the four key sectors in each of the three regions considered here. To define the model training runs, we use maximin Latin Hypercube sampling,

following (Lee et al., 2011), to select 60 distinct sets of emission changes in the range 0–120% to apply to the 12 sources considered here (see Table S2 in the supplementary material for the values used). We perform WRF-Chem model simulations for each of these 60 emission scenarios for the period 21–31 October and build emulators for daily $PM_{2.5}$ in Beijing for each day of the period (11 emulators in total) using the model output. The emulators are built using the DICE-Kriging package in R (Roustant et al., 2012), following the approach of Ryan et al. (2018). We validate the emulators using a leave-one-out approach,

successively rebuilding all 11 emulators using only 59 of the 60 model outputs and comparing them against the remaining one. The agreement between the model and emulator outputs results in an $R^2$ of 0.99 (see Figure S4) indicating that the emulators reproduce the model $PM_{2.5}$ concentrations very well.

We first demonstrate use of the emulators to investigate the contribution of each emission source to $PM_{2.5}$ in Beijing by performing global sensitivity analysis. This provides a robust method for identifying which sources most affect simulated

$PM_{2.5}$ concentrations. We calculate a global sensitivity index for each source which quantifies its contribution to the total variance in $PM_{2.5}$ across all combinations of changes in other sources (Lee et al., 2013; Ryan et al., 2018). We use the computationally efficient extended Fourier Amplitude Sensitivity Test (eFAST) algorithm (Saltelli et al., 1999) to compute these global sensitivity indices. The eFAST algorithm needs thousands of model evaluations from across the emission parameter space considered, and while this would be computationally prohibitive with the WRF-Chem model, it is relatively straight-

forward with the emulators. We conduct 10,000 runs with the emulators to calculate global sensitivity indices for each of the 12 emission sources. Since these sensitivity indices are based on the variance in $PM_{2.5}$, we take the square root to permit a direct comparison with the contributions calculated using one-at-a-time local sensitivity analysis as described in section 4. We find that the relative contributions from the different regions and sectors derived through these different approaches are generally very similar, see Figure S5. However, we note that power and transport sectors, which are the dominant sources

of precursor gases for secondary inorganic aerosol formation, show slightly higher contributions calculated this way than when derived from one-at-a-time source changes, reflecting the importance of interactions between these sectors for secondary aerosol formation. These results lend confidence in the use of the emulators and demonstrate the additional value that they provide.





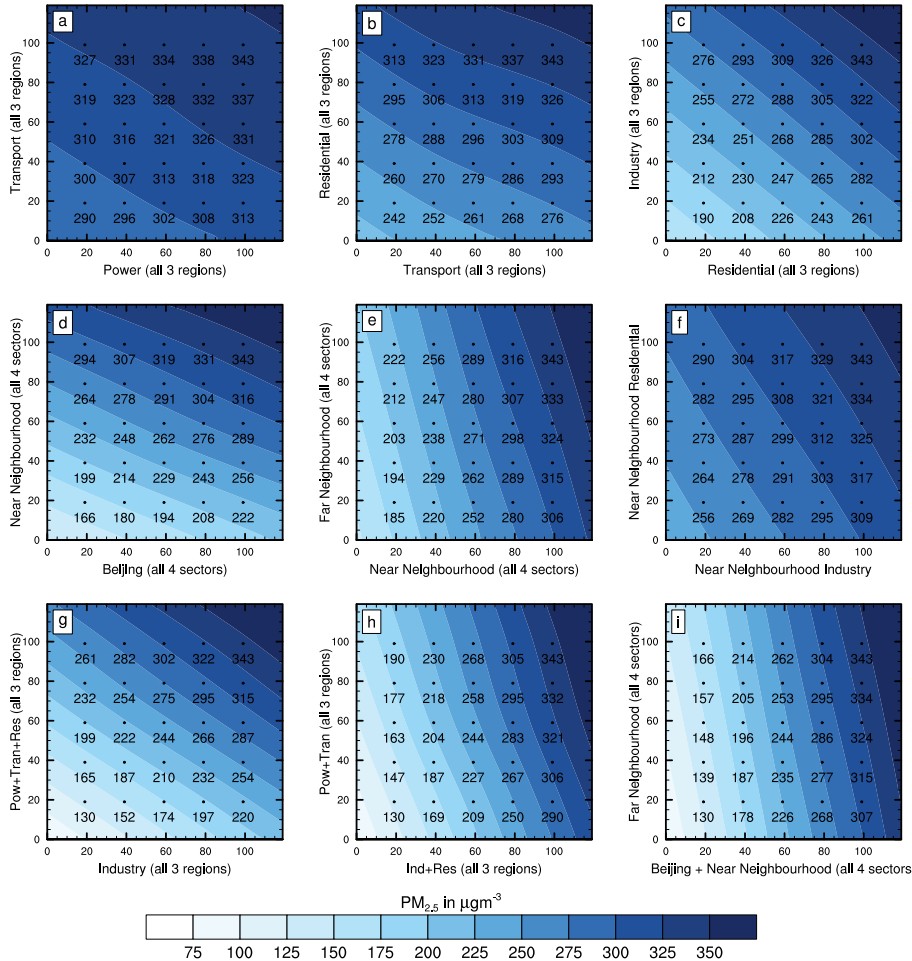

**Figure 10.** Response of daily mean $PM_{2.5}$ concentrations in Beijing (in $\mu g\,m^{-3}$) to changes in specific sectoral and regional emissions. Axes show the scaling applied to the relevant source (in %) starting on 21 October; contours show the corresponding daily mean $PM_{2.5}$ in Beijing on 24 October (concentrations at 20% intervals are labelled for clarity).

Concentration response surfaces were developed for daily average $PM_{2.5}$ in Beijing for each day of the period (21–31 October) using the emulators. Figure 10 presents response surfaces for some key combinations of emission sources for 24 October, the day with highest $PM_{2.5}$ concentrations. $PM_{2.5}$ responses to changes in key sectors across all source regions together are shown in Fig 10(a-c). Emission reductions in the transport sector are more efficient than those in the power sector in reducing total $PM_{2.5}$ over Beijing (Fig 10a); an 80% reduction in transport emissions results in a reduction of $30\,\mu g\,m^{-3}$ (from $343\,\mu g\,m^{-3}$ to $313\,\mu g\,m^{-3}$) while the same reduction in the power sector leads to a reduction of only $16\,\mu g\,m^{-3}$. Non-linearity in the $PM_{2.5}$ response is also evident here; for example, a reduction in the transport sector by 60% with no change in the power sector leads to a reduction in $PM_{2.5}$ of $20\,\mu g\,m^{-3}$ (343 to $323\,\mu g\,m^{-3}$) whereas the same reduction applied when the power sector is at 20% strength reduces $PM_{2.5}$ by $27\,\mu g\,m^{-3}$ (327 to $300\,\mu g\,m^{-3}$). These response surfaces demonstrate that





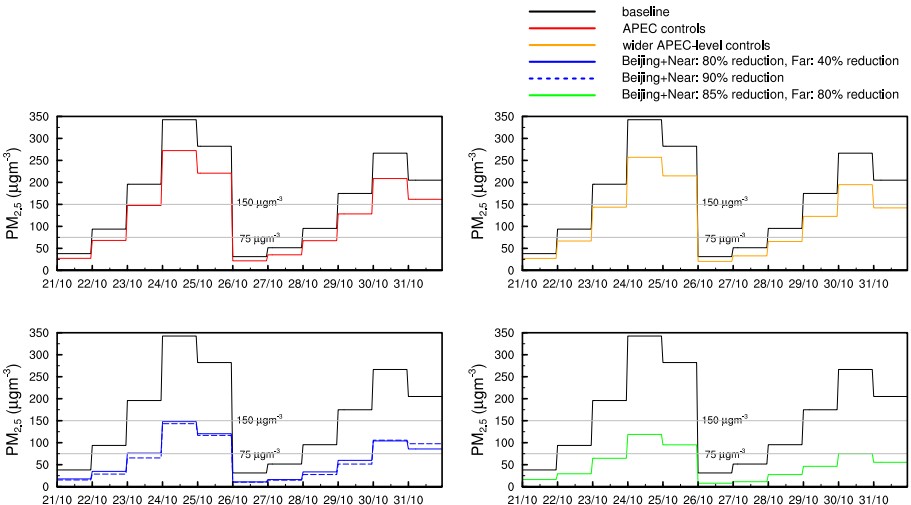

**Figure 11.** Average daily mean $PM_{2.5}$ in Beijing during October 2014 based on emission control scenarios including (a) controls on Beijing and near-neighbourhood sources applied during APEC, (b) APEC controls extended to the far-neighbourhood region, (c) controls needed to attain a standard of $150\,\mu g\,m^{-3}$ throughout the period, and (d) controls needed to meet a standard of $75\,\mu g\,m^{-3}$ on 30 October.

the industrial sector makes the greatest contribution to $PM_{2.5}$ in Beijing, closely followed by the residential sector, and that strong controls on these two sectors alone would bring $PM_{2.5}$ concentrations down towards $150\,\mu g\,m^{-3}$ under the conditions prevailing on this day.

Response surfaces for emission changes over different source regions with all sectors combined are shown in Fig 10(d-e). Near Neighbourhood sources make the greatest contributions to $PM_{2.5}$ in Beijing under these polluted conditions, with smaller contributions from Beijing and Far Neighbourhood sources. An 80% reduction in Near Neighbourhood sources alone would reduce $PM_{2.5}$ concentrations by $121\,\mu g\,m^{-3}$, and the dominant emissions sector in this region, industrial sources, is responsible for $53\,\mu g\,m^{-3}$ of this (Fig 10f). However, it is clear that reductions in specific regions or sectors alone are insufficient to bring $PM_{2.5}$ concentrations below $150\,\mu g\,m^{-3}$. We therefore show the effect of combined emission reductions in Fig 10(g-i). Reduction in emissions of 80% across all 12 sources bring $PM_{2.5}$ levels down to $130\,\mu g\,m^{-3}$, a drop of $213\,\mu g\,m^{-3}$. This remains insufficient to meet the national air quality standard of daily average $PM_{2.5}$ concentrations below $75\,\mu g\,m^{-3}$. Meeting this standard would require removal of all emission sources for all three regions which would bring down the levels to $79\,\mu g\,m^{-3}$. Greater reductions in $PM_{2.5}$ concentrations on 24 October would require emission controls to be applied one or two days earlier than the start of the period considered here, 21 October, as demonstrated in section 3. Nonetheless, we demonstrate that these response surfaces provide valuable guidance on the reductions in $PM_{2.5}$ in Beijing that are possible from different combinations of emission reductions without the need to perform additional air quality model simulations.

The response surfaces can be used to explore the effects of a wide range of possible emission control scenarios. We show examples of these for daily mean $PM_{2.5}$ in Beijing in Figure 11. For the emission reductions applied during the APEC period, we find reductions in $PM_{2.5}$ of about 23%, very similar to those derived from an independent model simulation using these emis-



sion changes (Ansari et al., 2019). Extending the same emission controls to a wider region over far neighbourhood provinces would lead to an additional improvement, giving a total reduction in $PM_{2.5}$ of 27%, but this remains insufficient to meet even the secondary air quality standard of $150\,\mu g\,m^{-3}$ on 24 October. Meeting this standard would require an emission reduction of 90% across all Beijing and Near-Neighbourhood sources. Alternatively, it could be met with an emission reduction of 80%

across all Beijing and Near-Neighbourhood sources along with a reduction of 40% across all Far-Neighbourhood sources. The choice of source reductions needed to meet a given standard can be taken directly from the corresponding response surface (see Figure 10). Meeting the primary air quality standard of $75\,\mu g\,m^{-3}$ on 30 October would be possible with a reduction of 85% across Beijing and Near-Neighbourhood sources and 80% across Far-Neighbourhood sources, see Figure 11. The standard would still be exceeded on 24 and 25 October under these conditions, but we note that our period of interest started on

21 October, only three days before this pollution episode. As shown earlier, emissions over more distant regions from 4-5 days previously affect Beijing during this pollution episode, and controls would have been needed before 21 October, perhaps over an even wider region than we consider here, to meet the standards over this full period.

We note that there have been considerable reductions in emissions since 2014 as a result of the Chinese government's clean air policies which have included ultra-low emission standards for power plants, phase-out of outdated industrial infrastructure,

replacement of coal with natural gas for residential heating, and higher fuel standards for vehicles. These have led to a significant improvement in air quality (Zhang et al., 2019; Zheng et al., 2018). Between 2014 and 2017, observed annual average $PM_{2.5}$ concentrations dropped from $100\,\mu g\,m^{-3}$ to $64\,\mu g\,m^{-3}$ in the Beijing-Tianjin-Hebei region (Wang et al., 2020) and from $86\,\mu g\,m^{-3}$ to $58\,\mu g\,m^{-3}$ in Beijing (Cheng et al., 2019). To explore the effect of recent emission changes, we apply emission reductions for Beijing (75% for industry and power sources, 80% for residential sources, and no change for traffic sources)

and for neighbouring provinces (40% for industry and power, 30% for residential sources, no change for traffic sources) that reflect recent emission assessments (Cheng et al., 2019; Zheng et al., 2018). We find that daily mean $PM_{2.5}$ concentrations in Beijing over the October period examined here are reduced by an average of 36%, close to the observed annual reduction of 33%. However, $PM_{2.5}$ concentrations still peak at $253\,\mu g\,m^{-3}$ and $175\,\mu g\,m^{-3}$ during the episodes of 24 and 30 October, respectively, and these levels remain well above the national daily air quality standard. Recent independent modelling studies

have highlighted the continued persistence of haze episodes in the region even after the implementation of clean air policies (Dang and Liao, 2019; Li et al., 2019). This emphasises that short-term emergency measures remain indispensable for avoiding the most severe haze episodes and our study provides a guiding framework for their formulation.

We carefully chose a period covering a diverse set of meteorological conditions for our temporal, sectoral and global sensitivity analyses (see Table S2 and Figure S2 in Ansari et al. (2019) for more details on meteorological conditions during the

study period), however our approaches are fully generalizable and can be applied to other longer periods and different seasons in order to build an even more general picture.





# 6 Conclusions

In this study we have investigated the temporal, regional and sectoral contributions to particulate air pollution in Beijing in autumn using a high-resolution regional air quality model with the aim of informing short-term emission control policies to mitigate major $PM_{2.5}$ pollution episodes. We find that the effects of local emission reductions in Beijing can be substantial, but

typically only contribute about 20% of $PM_{2.5}$ concentrations during pollution episodes, and the effects rarely extend beyond the day of emissions reduction. Controls on emissions in neighbouring regions over the North China Plain have a larger impact on $PM_{2.5}$ in Beijing, typically about 35% during pollution episodes, peaking the day following application of controls, and they can persist for a number of days. Emissions from more distant regions over North China can also have a substantial impact under some meteorological conditions, and the effect of emission changes are more dispersed and may persist over Beijing for

as much as five days. Using simple one-at-a-time sensitivity studies we identify industrial and residential emissions from near-neighbourhood regions of the North China Plain as the two largest sources of $PM_{2.5}$ in Beijing, although there is substantial variability in their contributions driven by meteorology, and on cleaner days under northwesterly flow $PM_{2.5}$ concentrations are principally from local Beijing sources, particularly from the residential and industrial sectors.

To provide a more comprehensive analysis of source contributions that accounts for the interactions between precursors from

different sectors and regions, we apply Gaussian Process emulation to derive response surfaces for daily $PM_{2.5}$ concentrations in Beijing and demonstrate how these may be used to guide choice of emission controls to meet specific air quality standards. Focusing on a polluted period in October 2014 we show that large emission reductions exceeding 80% may be needed to meet standards, but that the magnitude of the reductions depends on the regional extent of the controls, and on their timing. Controls focused on the North China Plain alone may be insufficient to keep air quality within the national standards, and

this provides strong support for the application of controls over a wider area such as those implemented in the recent "2+26" programme (Chen et al., 2019). While we have focused on the October-November 2014 period associated with the important APEC Summit to explore source sensitivities, we note that emissions are changing rapidly in both Beijing and surrounding regions (Cheng et al., 2019; Zheng et al., 2018) leading to spatial and temporal changes in air pollution (Xiao et al., 2020; Xu and Zhang, 2020). The temporal sensitivity and emulation approaches we have demonstrated here may nonetheless serve as

a valuable guide in policy formulation for Beijing during the autumn/winter period. More generally, these methods could be used to build a formal framework to guide policy decisions aimed at mitigating short-term air pollution episodes throughout the year, and may be equally valuable in other cities around the world.

*Code and data availability.* The WRF-Chem model code is available from http://www2.mmm.ucar.edu/wrf/users/download/. The model configuration and surface pollutant distributions generated in this study are available from the Lancaster University data archive at http:

//dx.doi.org/10.15125/XXXXXX.





*Author contributions.* TUA, OW and ZW designed this study, and TUA performed the model simulations, analysis and visualization. JL provided emissions data and expertise on the model configuration. ER generated code for emulating the model output. YC provided scientific expertise and critical support on model operation. TUA and OW prepared the manuscript with input from all coauthors.

*Competing interests.* The authors declare that they have no conflict of interest

5  *Acknowledgements.* Tabish Umar Ansari and Oliver Wild thank the UK Natural Environment Research Council for support under grants NE/N006925/1 and NE/N006976/1. Jie Li thanks the National Natural Science Foundation of China (grant nos. 41571130034 and 91744203). Tabish Umar Ansari acknowledges Dennis Shea, Adam Philips, Rashed Mahmood and Rick Brownrigg of the "ncl-talk" online forum for providing assistance on data visualization, and thanks the Lancaster Environment Centre for providing funding to carry out his PhD research.



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
