# Peer review of "Temporally-resolved sectoral and regional contributions to air pollution in Beijing: Informing short-term emission controls"

_Atmospheric Chemistry and Physics, 2020_

## Referee Comment (RC1) · Anonymous Referee #1 · 28 Sep 2020

**Review**

Atmospheric Chemistry and Physics

**Title**

Temporally–resolved sectoral and regional contributions to air pollution in Beijing: Informing short–term emission controls

**Authors**

Tabish Umar Ansari, Oliver Wild, Edmund Ryan, Ying Chen, Jie Li, and Zifa Wang

**Summary**

This study analysed the sectoral, regional, and temporal contributions to two air pollution episodes in October 2014 across Beijing, China. Chemical transport model simulations were used to determine the temporal, regional, and individual emission sector contributions to ambient fine particulate matter ($PM_{2.5}$) concentrations, in addition to training emulators to predict air quality based on multiple emission sector variations. The paper explored the impacts of various controls under different meteorological conditions and found the local and regional importance of residential and industrial emissions. The topic of this paper is relevant to the scope of Atmospheric Chemistry and Physics. The paper used a relatively novel approach to provide an interesting understanding of how short–term emission controls influence air pollution episodes in Beijing, China.

My main criticisms regard further discussions and clarifications.

The authors emphasise the importance of short–term emergency measures for air pollution episodes, whilst convincingly demonstrating their minor role relative to meteorology and their limited effectiveness even at stringent implementations. This dichotomy should be further discussed, relative to long–term emission reductions and their implications for public health. For example, the authors found key contributions from residential and industrial emissions, similar to previous studies, and mentioned the large impacts of recent long–term emission reductions in China, which mainly focused on industrial and power emissions (Zheng *et al* 2018). However, these previous long–term emission reductions did not explicitly control for residential emissions, despite their key importance to both ambient and household $PM_{2.5}$ exposure (Zhao *et al* 2018). Current policies in Beijing and surrounding municipalities aim to specifically address the largely neglected and substantial emissions from residential solid fuel use (National Development and Reform Commission of China 2017), with large potential public health benefits (Meng *et al* 2019). These are especially important considering that the risks to public health from air pollution exposure are significantly larger at longer time scales.

The authors mention that emissions in and around Beijing are under rapid change and that individual air pollution episodes are dependent on specific meteorological conditions. Hence, the generalisability of this framework for future air pollution episodes need to discussed.

Overall, this well–written paper provides an interesting application of a relatively novel method to important issues surrounding the control of substantial air pollution exposure. The paper would be improved from enhanced discussions and clarifications.

**Comments**

1. The authors should state the focus on *ambient* air quality, as China still experiences poor *household* air quality, which is confirmed by this studies finding of the importance of residential emissions.
2. Page 2 line 4, page 4 lines 6 and 10, page 11 lines 4, 6, and 7: Define acronyms at first use.
3. It would aid the reader to specify $PM_{2.5}$ *concentrations* or *emissions*, rather than using $PM_{2.5}$ alone (e.g. page 2 line 27, page 6 line 4, page 7 lines 28 and 29, Figure 4, page 7 line 1, Figure 6, page 9 line 12, and other instances).
4. Figure 8 and 11: The baseline daily–mean $PM_{2.5}$ concentrations are more than the sum of the local, near–neighbourhood, and far–neighbourhood sources. For example, after removing all emission sources for all three regions daily–mean $PM_{2.5}$ concentrations remain at 79 $\mu g\ m^{-3}$. It would be useful to discuss what is contributing to this remaining large exposure.
5. Section 5: Methods, evaluation, and results are combined. The clarity would be improved if these were separated.
6. Figure 8 and 11: Perceptually–uniform colour maps would improve the clarity of the Figures (e.g. viridis, ColorBrewer 2.0).
7. The paper has many figures, which may dilute key findings. Some of the figures could be moved to the Supplementary.
8. Page 16 lines 7–10: References needed.
9. Figure S2: Define D02 and D03.

**References**

Arriagada N B, Palmer A J, Bowman D M J S, Morgan G G, Jalaludin B B and Johnston F H 2020 Unprecedented smoke-related health burden associated with the 2019–20 bushfires in eastern Australia *Med. J. Aust.* 2019–20

GBD 2017 Risk Factor Collaborators 2018 Global, regional, and national comparative risk assessment of 84 behavioural, environmental and occupational, and metabolic risks or clusters of risks for 195 countries and territories, 1990–2017: a systematic analysis for the Global Burden of Disease Stu *Lancet* **392** 1923–94

Grange S K, Carslaw D C, Lewis A C, Boleti E and Hueglin C 2018 Random forest meteorological normalisation models for Swiss PM10 trend analysis *Atmos. Chem. Phys.* **18** 6223–39

Liu J, Kiesewetter G, Klimont Z, Cofala J, Heyes C, Schöpp W, Zhu T, Cao G, Gomez Sanabria A, Sander R, Guo F, Zhang Q, Nguyen B, Bertok I, Rafaj P and Amann M 2019 Mitigation pathways of air pollution from residential emissions in the Beijing-Tianjin-Hebei region in China *Environ. Int.* **125** 236–44

Meng W, Zhong Q, Chen Y, Shen H, Yun X, Smith K R, Li B, Liu J, Wang X, Ma J, Cheng H, Zeng E Y, Guan D, Russell A G and Tao S 2019 Energy and air pollution benefits of household fuel policies in northern China *Proc. Natl. Acad. Sci.* 201904182

Ministry of Environmental Protection of China 2017 Beijing-Tianjin-Hebei and surrounding areas 2017 Air Pollution Prevention and Control Work Plan *Gov. China* Online: http://dqhj.mee.gov.cn/dtxx/201703/t20170323_408663.shtml

National Development and Reform Commission of China 2017 Work plan for clean heating in winter in northern China (2017–2021) *Gov. China* Online: http://www.gov.cn/xinwen/2017-12/20/content_5248855.htm

Qin Y, Wagner F, Scovronick N, Peng W, Yang J, Zhu T, Smith K R and Mauzerall D L 2017 Air quality, health, and climate implications of China's synthetic natural gas development *Proc. Natl.*

*Acad. Sci.* **114** 4887–92

Venter Z S, Aunan K, Chowdhury S and Lelieveld J 2020 COVID-19 lockdowns cause global air pollution declines *Proc. Natl. Acad. Sci. U. S. A.* **117** 18984–90

Zhao B, Zheng H, Wang S, Smith K R, Lu X, Aunan K, Gu Y, Wang Y, Ding D, Xing J, Fu X, Yang X, Liou K-N and Hao J 2018 Change in household fuels dominates the decrease in PM2.5 exposure and premature mortality in China in 2005–2015 *Proc. Natl. Acad. Sci.* **115** 12401–6

Zheng B, Tong D, Li M, Liu F, Hong C, Geng G, Li H, Li X, Peng L, Qi J, Yan L, Zhang Y, Zhao H, Zheng Y, He K and Zhang Q 2018 Trends in China's anthropogenic emissions since 2010 as the consequence of clean air actions *Atmos. Chem. Phys.* **18** 14095–111

---

## Referee Comment (RC2) · Anonymous Referee #2 · 22 Oct 2020

**Review of 'Temporally-resolved sectoral and regional contributions to air pollution in Beijing: Informing short-term emission controls'**

**General comments**

This paper analyses the impact of various short-term emission controls on $PM_{2.5}$ concentrations in Beijing. Various aspects are analysed in multiple model experiments, including the timing of the emission control, the area of emissions control, the emissions sector that is controlled, and interactions between different controls. This is a strong paper containing a great deal of valuable analysis, and valuable insights into air pollution episode control policies. However, a more detailed description of the methods is needed to be able to understand whether the results have been interpreted correctly.

**Specific comments**

- In this paper you refer to the evaluation of your model setup in a previous paper. However, I think it is necessary to include the results of evaluation of these simulations against measurements within this paper. In some cases, in Ansari et al. (2019), the model showed biases for key pollutants. While this by no means invalidates the results of this study, the reader of this paper should be made aware of the biases in the model (and their direction), possible reasons for these issues and how they could affect the interpretation of your results. It would be particularly useful to know whether the model estimated the magnitude of the episodes correctly, which you could show by adding measurement data to Figure 1.
- On page 3, the sentence beginning on line 18 details the two phases of APEC emissions controls. Please cite the source of your information on the controls here.
- There is no mention of whether there was any spinup for the model runs. If there was no spinup, the PM concentration reductions achieved by each day of emissions cuts may be unrealistic. If the baseline run covers the 14-day period (should be specified), and there is no spinup for the 5 day runs, then emissions reductions may be overestimated. For example, in 'Run No. 10' the first day would be expected to have lower PM concentrations compared with baseline day 10 anyway due to it having a 'cold start.' Please specify whether spinups were performed, if 5 day runs were initialised with fields from the baseline, or whether they were cold starts.
- If I have interpreted it correctly, to make Figure 3 you calculate the difference between the $PM_{2.5}$ concentrations in the baseline run, and in the runs with a day of reduced emissions. So the height of the stacked line shows the sum of the reductions in PM made by implementing the control in each individual run. However, since the graph appears to be a time series of PM, the first impression on seeing this figure is that you portioned the total PM by the day on which it was emitted. However, due to the non-linearity of the relationship between PM (and its precursors) emission volume and PM concentrations, and due to the lingering and transport of PM, the concentration reductions sum do not account for the total PM. I.e. on the 24[th], the sum of reductions is around 120 ug/m3, whereas Figure 1 shows the daily mean was over 350 ug/m3. I suggest the figure should be adjusted so it is clearer that it represents the concentration reduction as a result of emissions reductions for each day. You could do this changing the y axis to 'concentration reduction,' which would flip the graph horizontally. Another issue is that Figure 3 suggest that these would be the reductions achieved by a combination of emission reductions on those days (while the simulations are actually separate so will not simulate any synergistic effects). A better way to represent this, while making your experimental design clearer, could be to us e a format similar to Figure S1, with each emission reduction shown separately.
- It would be helpful to define how you calculate the integrated contribution so that the unit of 'µg m$^{-3}$h' (should it be 'µg m$^{-3}$h$^{\textbf{-1}}$'?) can be understood.

- Multiple source apportionment studies suggest that agriculture and biomass burning are major contributors to $PM_{2.5}$-caused mortality in China, with similar contributions to transport and power generation sectors. It should be specified whether additional run that estimates the background contribution includes these sectors, and whether their emissions are included in the model at all.
- It would be useful to include a breakdown of emission volumes and spatial pattern by sector so that the concentrations reductions can be put in context. For example, controls on the power sector seem to produce a relatively small PM decrease, but is this due to this sector comprising a relatively small proportion of emissions, the composition of species emitted from power sector, or the spatial distribution of emissions?
- Please could you comment on the sources of the 'background' levels of $PM_{2.5}$ (i.e. natural + outside domain). This is especially relevant for understanding the implications of the sentence on P15L10, where you report that even with no emissions, $PM_{2.5}$ concentrations would be 79 ug m$^{-3}$ under these weather conditions.

**Technical Corrections**

P1L1 – 'to air pollution' rather than 'to air quality' sounds better in this sentence and is consistent with the title.

For your date axes on Figures – sometimes they look a bit crowded (such as figure 7). I suggest changing the axis title to 'day in October' and removing the '/10' from the tick labels.

Figure 11 - missing subplot labels. Perhaps the subplots in this figure could be merged into one large figure which would make it easier to visually compare the runs.

---

## Author Comment (AC1) · 22 Jan 2021

**Authors' reponse to reviewers' comments**
(The reviewer's comments are in black while authors' responses are shown in blue)

**Reviewer 1**

Summary
This study analysed the sectoral, regional, and temporal contributions to two air pollution episodes in October 2014 across Beijing, China. Chemical transport model simulations were used to determine the temporal, regional, and individual emission sector contributions to ambient fine particulate matter (PM 2.5 ) concentrations, in addition to training emulators to predict air quality based on multiple emission sector variations. The paper explored the impacts of various controls under different meteorological conditions and found the local and regional importance of residential and industrial emissions. The topic of this paper is relevant to the scope of Atmospheric Chemistry and Physics. The paper used a relatively novel approach to provide an interesting understanding of how short-term emission controls influence air pollution episodes in Beijing, China.

My main criticisms regard further discussions and clarifications.

The authors emphasise the importance of short-term emergency measures for air pollution episodes, whilst convincingly demonstrating their minor role relative to meteorology and their limited effectiveness even at stringent implementations. This dichotomy should be further discussed, relative to long-term emission reductions and their implications for public health. For example, the authors found key contributions from residential and industrial emissions, similar to previous studies, and mentioned the large impacts of recent long-term emission reductions in China, which mainly focused on industrial and power emissions (Zheng et al., 2018). However, these previous long-term emission reductions did not explicitly control for residential emissions, despite their key importance to both ambient and household PM 2.5 exposure (Zhao et al 2018). Current policies in Beijing and surrounding municipalities aim to specifically address the largely neglected and substantial emissions from residential solid fuel use (National Development and Reform Commission of China 2017), with large potential public health benefits (Meng et al 2019). These are especially important considering that the risks to public health from air pollution exposure are significantly larger at longer time scales.

The authors mention that emissions in and around Beijing are under rapid change and that individual air pollution episodes are dependent on specific meteorological conditions. Hence, the generalisability of this framework for future air pollution episodes need to discussed.

Overall, this well-written paper provides an interesting application of a relatively novel method to important issues surrounding the control of substantial air pollution exposure. The paper would be improved from enhanced discussions and clarifications.

We thank the reviewer for their positive comments about our paper. We agree that the future pollution mitigation strategies in China would benefit from further reducing residential emissions which would improve both ambient and household air quality. However, our focus in this study is on better understanding large-scale source contributions to ambient air pollution in Beijing to facilitate the design of optimal short-term emission control strategies. Major episodes with poor air quality still occur despite the overall long-term emission reductions in place. Our unique focus here is specifically on short-term controls, in contrast to the wealth of studies in the literature on long-term controls. While our studies are set in the context of 2014 conditions to address particular episodes that were of interest for the APEC summit period, the approaches are fully generalizable to other periods with appropriate underlying emissions conditions.

While meteorological conditions can have a larger impact on severe pollution episodes than short-term emission controls, we show that the latter can still be important in reducing pollution levels under these conditions. This resolves the apparent dichotomy alluded to by the reviewer.

We highlight that although our sectoral analysis focuses principally on two major episodes in Oct 2014, we explored a longer period spanning 28 days from 18 Oct-15 Nov (described on page 5, lines 25-27; see also Figure S1 and Table S1) which covered a diverse range of meteorological conditions. The range of conditions covered is illustrated in Figure 5. November was a cleaner period when emission controls were applied for APEC, and there were few poor air quality episodes, as also shown in our previous study (Ansari et al., 2019, ACP) where more detailed analysis of key meteorological parameters during the entire period which can be found in Table S2 and Figure S2. We have now added a sentence to discuss how approaches used are generalizable (page 16 lines 28-31).

Specific comments and their responses:

1. The authors should state the focus on ambient air quality, as China still experiences poor household air quality, which is confirmed by this studies finding of the importance of residential emissions.

We have now added the word "ambient" (e.g., on Page 2 Line 4) to emphasize that the focus of the study is on outdoor air pollution.

2. Page 2 line 4, page 4 lines 6 and 10, page 11 lines 4, 6, and 7: Define acronyms at first use.

We have now fully defined these acronyms at all these instances.

3. It would aid the reader to specify PM 2.5 concentrations or emissions, rather than using PM 2.5 alone (e.g. page 2 line 27, page 6 line 4, page 7 lines 28 and 29, Figure 4, page 7 line 1, Figure 6, page 9 line 12, and other instances).

We have now replaced "$PM_{2.5}$" with "$PM_{2.5}$ concentrations" at all these locations to make it clear that we refer to concentrations rather than emissions.

4. Figure 8 and 11: The baseline daily−mean PM 2.5 concentrations are more than the sum of the local, near−neighbourhood, and far−neighbourhood sources. For example, after removing all emission sources for all three regions daily−mean PM 2.5 concentrations remain at 79 $\mu$g m −3 . It would be useful to discuss what is contributing to this remaining large exposure.

The reviewer rightly points out that the daily mean $PM_{2.5}$ concentrations are higher than the sum of the individual contributions from local, near−neighbourhood, and far−neighbourhood sources. The differences are due to the contributions from background and natural sources, and contributions from emissions before the start of the simulations. The contributions from these sources are clearly quantified in the previous section and in particular in Figure 7, which shows both the effect of background sources and the interaction of different sources with each other. The is already mentioned in the text on page 10 Lines 11-12, and we have added another sentence to make this clearer (page 11 lines 1-3).

5. Section 5: Methods, evaluation, and results are combined. The clarity would be improved if these were separated.

The evaluation of the emulation approach has been reduced here, and the figure comparing Global Sensitivity Indices with the one-at-a-time contributions has been moved to the supplement. This simplifies the structure of the section so that it now focusses principally on the results of applying emulation, following a brief introduction that distinguishes the approaches from those used in the earlier sections

6. Figure 8 and 11: Perceptually–uniform colour maps would improve the clarity of the Figures (e.g. viridis, ColorBrewer 2.0).

Figures 8 and 11 (and S6 in supplement) have now been replotted using a perceptually uniform colour palette to improve accessibility for visually-impaired readers.

7. The paper has many figures, which may dilute key findings. Some of the figures could be moved to the Supplementary.

Figure 10 comparing the use of emulation for quantifying source contributions with that derived from one-at-a-time source perturbations has now been moved to the supplement as Figure S5. The remaining figures show key results and are needed to understand the approach, results and conclusions of the paper.

8. Page 16 lines 7–10: References needed.

References have now been added here as requested.

9. Figure S2: Define D02 and D03.

D02 and D03 have now been defined in the caption.

**Reviewer 2**

General comments
This paper analyses the impact of various short-term emission controls on PM 2.5 concentrations in Beijing. Various aspects are analysed in multiple model experiments, including the timing of the emission control, the area of emissions control, the emissions sector that is controlled, and interactions between different controls. This is a strong paper containing a great deal of valuable analysis, and valuable insights into air pollution episode control policies. However, a more detailed description of the methods is needed to be able to understand whether the results have been interpreted correctly.

Thank you for your positive comments about our study. We have now included more details on the methods to make them clearer to the reader. Responses to specific comments are shown below:

Specific comments

1. In this paper you refer to the evaluation of your model setup in a previous paper. However, I think it is necessary to include the results of evaluation of these simulations against measurements within this paper. In some cases, in Ansari et al. (2019), the model showed biases for key pollutants. While this by no means invalidates the results of this study, the reader of this paper should be made aware of the biases in the model (and their direction), possible reasons for these issues and how they could affect the interpretation of your results. It would be particularly useful to know whether the model estimated the magnitude of the episodes correctly, which you could show by adding measurement data to Figure 1.

Figure 1 has now been updated to include observations. Further details on model performance against observations and the reasons for model bias have been added on Page 3 Lines 12-16.

2. On page 3, the sentence beginning on line 18 details the two phases of APEC emissions controls. Please cite the source of your information on the controls here.

Two relevant references (Wen et al., 2016 and Li et al., 2017) have now been added.

3. There is no mention of whether there was any spinup for the model runs. If there was no spinup, the PM concentration reductions achieved by each day of emissions cuts may be unrealistic. If the baseline run covers the 14-day period (should be specified), and there is no spinup for the 5 day runs, then emissions reductions may be overestimated. For example, in 'Run No. 10' the first day would be expected to have lower PM concentrations compared with baseline day 10 anyway due to it having a 'cold start.' Please specify whether spinups were performed, if 5 day runs were initialised with fields from the baseline, or whether they were cold starts.

The baseline run was 41 days long (10 October 0000hrs UTC – 19 November 2300hrs UTC) of which the first 9 days hours were excluded from the analysis here. All sensitivity runs had 'hot starts' initialized from the baseline restart files, so a substantial spin-up period was not needed. These sensitivity runs were started 16 hours before the start date in Beijing local time to account for the time zone and the emission changes were implemented from 00 hrs local time. We have now included these details at Page 5 Lines 30-31, Page 9 Lines 22-24 and Page 13 Line 12.

4. If I have interpreted it correctly, to make Figure 3 you calculate the difference between the PM 2.5 concentrations in the baseline run, and in the runs with a day of reduced emissions. So the height of the stacked line shows the sum of the reductions in PM made by implementing the control in each individual run. However, since the graph appears to be a time series of PM, the first impression on seeing this figure is that you portioned the total PM by the day on which it was emitted. However, due to the non-linearity of the relationship between PM (and its precursors) emission volume and PM concentrations, and due to the lingering and transport of PM, the concentration reductions sum do not account for the total PM. I.e. on the 24 th , the sum of reductions is around 120 ug/m3, whereas Figure 1 shows the daily mean was over 350 ug/m3. I suggest the figure should be adjusted so it is clearer that it represents the concentration reduction as a result of emissions reductions for each day. You could do this changing the y axis to 'concentration reduction,' which would flip the graph horizontally. Another issue is that Figure 3 suggest that these would be the reductions achieved by a combination of emission reductions on those days (while the simulations are actually separate so will not simulate any synergistic effects). A better way to represent this, while making your experimental design clearer, could be to us e a format similar to Figure S1, with each emission reduction shown separately.

Figure 3 shows the temporal contributions to $PM_{2.5}$ concentrations in Beijing from emission changes on successive days. These are indeed reductions in $PM_{2.5}$ concentrations, but we present them in a positive manner so that it is more intuitive for the reader. We have specifically chosen to present the contributions in a cumulative manner (rather than independently as the reviewer suggests) to emphasise that they build on each other and to highlight that the total effect on a single day is built up of contributions from a number of days of emissions. However, we acknowledge that this is not entirely clear from the caption, so have rewritten the caption to emphasise the cumulative nature of the contributions in particular.

The reviewer notes that the sum of the contributions here is substantially less than that shown in Figure 3. This is partly because it shows contributions from 30-50% emissions reductions as applied during APEC (as explained in the text) and partly because it reflects anthropogenic sources from a limited number of days.  The figure does not represent a source attribution, and this should now be clearer from the revised caption.

5. It would be helpful to define how you calculate the integrated contribution so that the unit of 'μg m -3 h' (should it be 'μg m -3 h -1 '?) can be understood.

Integrated contributions were calculated by computing the area under the curve of each 'pulse', which was derived by subtracting each 5-day sensitivity run from the baseline run for those five days. Since the y-axis of the pulse represents concentration in $\mu g\ m^{-3}$ and the x-axis time in hours, the area under the curve is dimensionally $\mu g\ m^{-3}\ h$. We have now included this information in Table S1.

6. Multiple source apportionment studies suggest that agriculture and biomass burning are major contributors to PM 2.5 -caused mortality in China, with similar contributions to transport and power generation sectors. It should be specified whether additional run that estimates the background contribution includes these sectors, and whether their emissions are included in the model at all.

The baseline run and all sectoral sensitivity runs included biogenic, biomass burning and agricultural emissions. The sizes of the coloured bars plotted in Figure 7 were calculated by subtracting the daily mean $PM_{2.5}$ concentrations in Beijing in the corresponding sectoral sensitivity run from those in the baseline run, and therefore the contributions from all other sources were eliminated. The additional run with all 12 sources removed allows us to extract the contributions from these sources along with the interactions between them. Subtracting the individual contributions from the 12 sources from this leaves us with the interactions, shown as the grey bars. The white bars show the remaining difference between this and the baseline run, which includes contributions from agricultural emissions, biomass burning emissions, biogenic emissions and background anthropogenic emissions from sources outside of the Far-Neighbourhood region.

We have now amended the text to note the inclusion of agricultural, biomass burning and biogenic emissions at Page 3 Lines 4-6. In addition, we have now clarified the meaning of 'background contribution' at Page 9 Lines 21-22.

7. It would be useful to include a breakdown of emission volumes and spatial pattern by sector so that the concentrations reductions can be put in context. For example, controls on the power sector seem to produce a relatively small PM decrease, but is this due to this sector comprising a relatively small proportion of emissions, the composition of species emitted from power sector, or the spatial distribution of emissions?

We have now included emission maps of CO, NO, $SO_2$, NMVOCs and $PM_{2.5}$ for industrial, power generation, transportation and residential sectors for model domain 2 in the supplement. Please see figures S7 – S8. The agriculture emissions sector only comprised $NH_3$ emissions, which are also included.

8. Please could you comment on the sources of the 'background' levels of PM 2.5 (i.e. natural + outside domain). This is especially relevant for understanding the implications of the sentence on P15L10, where you report that even with no emissions, PM 2.5 concentrations would be 79 ug m -3 under these weather conditions.

We have now added a clear definition of 'backgound contribution' at Page 9 Lines 21-22 (please refer to the response to comment 6 above). We have also altered the statement at Page 15 Line 2 which says "...removal of all emissions **from these sectors**..." which indicates that the remaining 79 ug m -3 $PM_{2.5}$ comes from background contribution.

Technical Corrections
P1L1 – 'to air pollution' rather than 'to air quality' sounds better in this sentence and is consistent with the title.
Thanks for pointing this out. We have now changed the statement accordingly.

For your date axes on Figures – sometimes they look a bit crowded (such as figure 7). I suggest changing the axis title to 'day in October' and removing the '/10' from the tick labels.
Figure 7 has now been updated with clearer tick labels

Figure 11 - missing subplot labels. Perhaps the subplots in this figure could be merged into one large
figure which would make it easier to visually compare the runs.
Thanks for this suggestion. All the scenarios are now overlaid on a single plot for better comparison.